

# Topological Data Analysis and Machine Learning for Recognizing Atmospheric River Patterns in Large Climate Datasets

Grzegorz Muszynski[1,2], Karthik Kashinath[2], Vitaliy Kurlin[1], Michael Wehner[2], and Prabhat[2]

[1]Department of Computer Science, University of Liverpool, Liverpool, L69 3BX, United Kingdom
[2]Lawrence Berkeley National Laboratory, Berkeley, California, 94720, United States

*Correspondence to:* Grzegorz Muszynski (gmuszynski@lbl.gov)

**Abstract.** Identifying weather patterns that frequently lead to extreme weather events is a crucial first step in understanding how they may vary under different climate change scenarios. Here we propose an automated method for recognizing atmospheric rivers (ARs) in climate data using topological data analysis and machine learning. The method provides useful information about topological features (shape characteristics) and statistics of ARs. We illustrate this method by applying it to outputs of version 5.1 of the Community Atmosphere Model (CAM5.1) and reanalysis product of the second Modern-Era Retrospective Analysis for Research & Applications (MERRA-2). An advantage of the proposed method is that it is threshold-free. Hence this method may be useful in evaluating model biases in calculating AR statistics. Further, the method can be applied to different climate scenarios without tuning since it does not rely on threshold conditions. We show that the method is suitable for rapidly analyzing large amounts of climate model and reanalysis output data.



# 1 Introduction

The importance of understanding of the behavior of extreme weather events in a changing climate cannot be overstated. A first step towards this challenging goal is to identify extreme events in large datasets. Identifying such events remains an important challenge for the climate science community for the following reasons:

- The identification process is critical in calculating statistics, including the frequency, location and intensity, of extreme weather events under different climate change scenarios.

- It is the first step in evaluating how well a climate model captures physical features of extreme events and characterizing their changes under global warming.

- As high performance computational technology continues to advance, there is an ever-increasing amount of data from
climate model output, reanalysis products and observations that demands rapid and automated detection and characterization of extreme events.

This study is part of ongoing efforts to provide automated methods that are able to identify extreme weather and climate events in large climate datasets (Prabhat et al., 2015; Ullrich and Zarzycki, 2017; Shields et al., 2018).

Extreme precipitation events in mid-latitudes are often associated with atmospheric rivers (ARs). Since the early 1990s, there
has been a growing interest in studying ARs (Zhu and Newell, 1994). ARs are long and narrow filaments of high concentrated water vapour in the lower troposphere. They are responsible for more than $\sim 90\%$ of the total poleward water vapour transport outside of the tropics (Newell et al., 1992; Newell and Zhu, 1994; Zhu and Newell, 1998). Most of them are associated with extreme winter storms and heavy precipitation events on the western coast of North America (Dettinger et al., 2011) and along the Atlantic European coasts (Fragoso et al., 2012; Lavers and Villarini, 2013). Due to the large amount of water that can
be transported by a single AR, they are potentially of high risk to society and often cause extreme flooding or have other devastating impacts when they make landfall (Ralph and Dettinger, 2011; Dettinger and Ingram, 2013; Ralph et al., 2016). On the other hand, ARs are critical in contributing to mountain snowpack and refilling reservoirs, thus mitigating drought, in areas such as the western United States, as in California (Guan et al., 2010; Dettinger, 2013). Figure 1 shows two examples of simulated ARs that deposit large amounts of rainfall on California and Washington state.

The first challenge in extreme event detection is to construct a quantitative definition of the event (Ullrich and Zarzycki, 2017). Once properly defined, developing a scheme to identify and track events in time and space can proceed. The AMS glossary defines an AR as follows, *"A long, narrow, and transient corridor of strong horizontal water vapor transport that is typically associated with a low-level jet stream ahead of the cold front of an extratropical cyclone. The water vapor in atmospheric rivers is supplied by tropical and/or extratropical moisture sources. Atmospheric rivers frequently lead to heavy*
*precipitation where they are forced upward—for example, by mountains or by ascent in the warm conveyor belt. Horizontal water vapor transport in the midlatitudes occurs primarily in atmospheric rivers and is focused in the lower troposphere"* (AMS, 2018). Note that this definition is qualitative and numerous methods have been proposed to make this quantitative and use them to detect ARs in regional and global climate data (Sellars et al., 2017), but none of these are free from a subjective





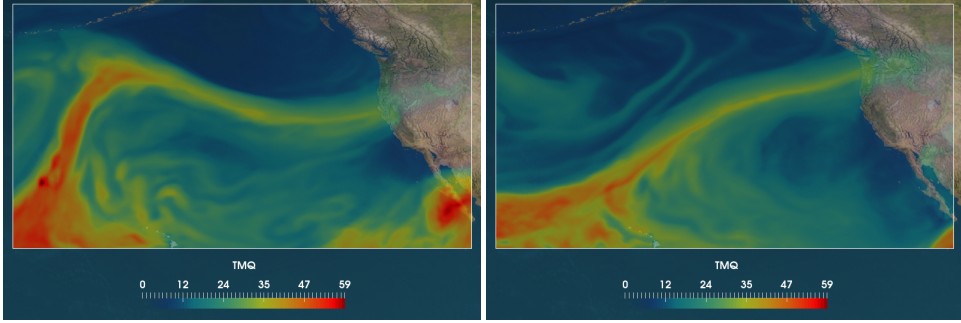

**Figure 1.** Sample images of atmospheric rivers - long filamentary structures reaching the West Coast of the United States (**Left:** landfall in a region of Northern California; **Right:** landfall in a region of Washington state). Shown is integrated water vapor ($\mathrm{kg\,m^{-2}}$) from a simulation of $5.1$ version of the Community Atmosphere Model (CAM5.1).

thresholding of some physical variable. Many existing techniques that have been designed for objective detection of ARs are based on a fixed threshold of more than $20\ \mathrm{kg\,m^{-2}}$ of Integrated Water Vapour (*IWV*) in the atmospheric column (Ralph et al., 2004) or more than $750\ \mathrm{kg\,m^{-1}\,s^{-1}}$ of Integrated Water Vapor Transport (IVT) (Sellars et al., 2017). Selecting appropriate threshold values of *IWV* or *IVT* in various climate scenarios remains an open challenge (Shields et al., 2018).

5    Some recent efforts focus on alternative approaches to characterize and detect extreme events, such as deep learning methods for pattern recognition (Liu et al., 2016; Racah et al., 2017), which use underlying features of datasets. In particular, the inherent design of these methods circumvent a critical challenge of event detection schemes, choosing suitable thresholds for different variables.

In this paper, we present an alternative approach to AR pattern recognition based on *topological data analysis* (TDA) (Ghrist, 2008; Carlsson, 2009, 2014) and *machine learning* (ML) (Kubat, 2015). Our approach uses TDA as a first step, followed by training a ML model to perform binary classification. TDA provides feature extraction tools using techniques from topology and computer science to study topological features of data (Carlsson, 2009, 2014). Topological features provide a unique and threshold-free way of describing crucial shape characteristics of physical phenomena, including weather events, in large datasets. We use a particular type of topological feature descriptors called *connected regions* (Edelsbrunner and Harer, 2010), which are obtained from scalar fields on a latitude-longitude grid (see Figure 2; Stage 1). The descriptors from positive and negative examples, *i.e.* events that are ARs and those that are not ARs, are then used in training a Support Vector Machine (SVM) classifier (Cortes and Vapnik, 1995; Chang and Lin, 2011), which is a ML model used for binary classification. In summary, the feature descriptors extract relevant topological information from a given scalar field, which is then used for training the ML classifier to perform the task of binary classification (see Figure 2; Stage 2). To the best of our knowledge, this is the first framework based on TDA and ML that has been introduced for recognizing weather patterns in large climate datasets. In this study, we focus on ARs making landfall along the west coast of North America, but the method is easily extendable to other regions.



The key contributions of this paper are: (i) We propose a novel method to identify ARs that is free from threshold selection; and (ii) We show that the framework of using TDA to extract topological feature descriptors and a ML classifier (SVM) provides high accuracy in recognizing AR patterns in both climate model output and reanalysis datasets across a range of spatial and temporal resolutions.

The rest of the paper is organized as follows: Section 2 describes datasets, the topological feature descriptors of ARs and non-ARs, the TDA algorithm and SVM classifier in more detail; Section 3 shows the results obtained with discussion; and Section 4 presents conclusions and future work.

## 2   Data and Method

### 2.1   Data

In this study, we use both climate model simulation output generated by version 5.1 of the Community Atmosphere Model [1] (CAM5.1) (Eaton, 2011) and a reanalysis product from the second Modern-Era Retrospective Analysis for Research & Applications [2] (MERRA-2) (Gelaro et al., 2017), respectively.

    The CAM5.1 climate model output is available at 25 km, 100 km, 200 km spatial resolutions, and both 3-hourly and daily temporal resolutions, for the period of January 1979 - December 2005 (Wehner et al., 2014). The MERRA-2 reanalysis data are

at 50 km spatial resolution and 3-hourly temporal resolution (instantaneous snapshots), for the period of January 1980 - June 2017. A summary of datasets is listed in Table 1. For both the CAM5.1 output and MERRA-2 data we use the total column Integrated Water Vapor, $IWV$ [3]. Note that this analysis could be performed on other relevant variables, including Integrated water Vapor Transport, $IVT$, the vertically integrated vector product of wind and water vapor, which is another commonly used variable for identifying ARs (Sellars et al., 2017). However, we note that $IWV$ is observable by satellite whereas $IVT$ is not.

Although outside the scope of this paper, an AR identification algorithm based on $IWV$ could offer an objective metric for evaluating both reanalysis products and climate models against observational data. We choose to use both 3-hourly and daily data because we anticipate the daily averages to smear out certain physical features of ARs. Further, 3-hourly data provides more event images labeled as ARs, which is useful for training in the machine learning model [4].

    Training a machine learning classifier, such as a Support Vector Machine (SVM) (see Subsection 2.2.2) requires labeled data

of events that are atmospheric rivers (ARs) and those that are not (non-ARs). In other words, each time step (snapshot) has to be tagged with a positive label (1 - *if it contains an AR*) or a negative label (0 - *if it does not contain an AR*). We use the parallel Toolkit for Extreme Climate events Analysis (TECA) (Prabhat et al., 2015) to obtain labels for training. The toolkit uses fixed threshold-based criteria (Ralph et al., 2004) to determine if there is AR in the given snapshot or not. The labels have been generated to for each dataset listed in Table 1. It is assumed that labels provided by TECA is "ground truth".

---

[1]CAM5.1 data are provided by the Lawrence Berkeley National Laboratory (LBNL), Berkeley; National Energy Research Scientific Computing Center.

[2]MERRA-2 data are provided by the University of California, San Diego (UCSD); Center for Western Weather and Water Extremes (CW3E).

[3]For the CAM5.1 this variable is called *TMQ*. For the MERRA-2 reanalysis data, it is called *IWV*. It is also called *prw* in the CF protocols.

[4]ML models tend to perform better when they have more training data.



**Table 1.** Data sources (climate model and reanalysis datasets).

| Climate Model | Period | Temporal Resolution | Spatial Resolution |
|---|---|---|---|
| CAM5.1 (historical run) | 1979-2005 | 3-hourly and daily | 25 km |
| CAM5.1 (historical run) | 1979-2005 | 3-hourly and daily | 100 km |
| CAM5.1 (historical run) | 1979-2005 | 3-hourly and daily | 200 km |
| MERRA-2 (reanalysis product) | 1980-2017 | 3-hourly | 50 km |

## 2.2 Atmospheric River Pattern Recognition Method

This subsection describes the 2 stages of the atmospheric river pattern recognition method (see Figure 2) based on *topological data analysis* (TDA) (Carlsson, 2009, 2014) and *machine learning* (ML) (Kubat, 2015):

- **Stage 1**: This stage applies techniques from topology [5] and algorithms from TDA to automatically infer relevant topological features from complex climate data including climate model output and reanalysis products. The TDA algorithm is based on the *Union-Find* data structure (Hopcroft and Ullman, 1973; Tarjan, 1975), which extracts *topological feature descriptors* of weather patterns, *i.e*, features of atmospheric rivers (ARs) and non-atmospheric rivers (non-ARs)), in a *threshold-free* way. These topological feature descriptors are called *connected regions* (Edelsbrunner and Harer, 2010) and are obtained from snapshots of global images on a latitude-longitude grid. The topological feature descriptors are provided as the input for the ML classifier in Stage 2.

- **Stage 2**: In this stage, a binary classification task is performed using the ML classifier, called Support Vector Machine (SVM) (Cortes and Vapnik, 1995; Chang and Lin, 2011). The classification task consists of two steps: i) Training the classifier to distinguish ARs from other weather events in the snapshots; and ii) Testing the constructed SVM model on the unlabeled descriptors to separate events into two groups (*i.e.*, ARs and non-ARs). The training process uses the topological feature descriptors (from Stage 1) and the ground truth labels (see Section 2.1) provided by TECA (Prabhat et al., 2015). The classifier performance is evaluated in the terms of accuracy, precision and sensitivity (see Section 2.3).

### 2.2.1 Stage 1: Topological Feature Descriptors of ARs and non-ARs

The aim of this stage is to automatically characterize AR and non-AR events in raw climate data. Most existing methods have been designed to use thresholds for identification of ARs (Shields et al., 2018). In contrast, the approach proposed here is *threshold-free* by employing *topological feature descriptors*, and in particular, *connected regions*. This approach is a type of TDA that is inspired by *persistence*, which is a concept in applied topology that summarizes topological variations across all values of the scalar field under consideration(Ghrist, 2008; Edelsbrunner and Morozov, 2012; Carlsson, 2009, 2014).

---

[5]*Topology* is the branch of mathematics studying properties of geometric objects (*e.g.* 2D grids) that are preserved under continuous deformations.



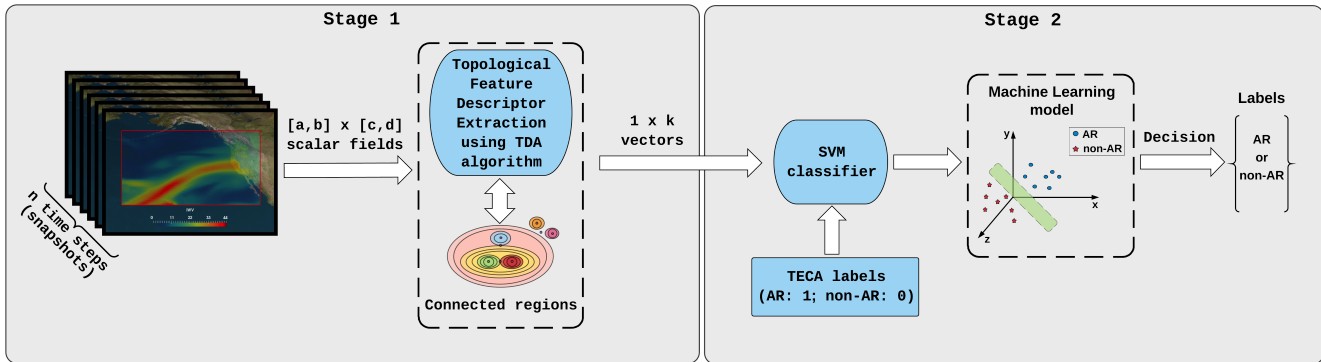

**Figure 2.** Block diagram of the AR pattern recognition method. *Stage* 1: The input of the method is a set of scalar fields ($n$ snapshots) of size $[a,b] \times [c,d]$ on the latitude-longitude grid, where $a$, $b$, $c$ and $d$ are the dimensions of grid spatial extent in real world. The topological data analysis algorithm (TDA) extracts connected regions from the snapshots of global images on the grid. The algorithm maintains the connected regions by varying IWV/TMQ values and dynamically keeps track of the regions in the grid. The connected regions are used as topological feature descriptors that characterize weather patterns (*i.e.*, atmospheric rivers (ARs) and non-atmospheric rivers (non-ARs)). The descriptors are $1 \times k$ vectors, where $k$ is greater than the maximal value of IWV/TMQ in a given set of scalar fields (in this case, k=60). *Stage* 2: The vectors are stacked on top of each other to form a $n \times k$ matrix and are fed into Support Vector Machine (SVM) classifier along with ground truth labels (*i.e.*, AR: 1 and non-AR: 0) provided by TECA (Prabhat et al., 2015). Finally, the SVM finds a suitable hyperplane (the green surface shown in the figure) that can cleanly separate events into two groups (*i.e.*, ARs and non-ARs). The output of the method is a set of $n$ labels based on the decision made by the SVM classifier.

Climate model output or reanalysis data may be represented as a mapping from the grid to a set of real values, which in our case is the *IWV* over $[0, L]$, where $L$ is the maximal value of the variable (here $L = 60 \text{ kg m}^{-2}$). It can be defined as follows

$$f : [a,b] \times [c,d] \rightarrow [0, L], \tag{1}$$

where $a$, $b$, $c$ and $d$ are the dimensions of the grid.

5     Every node (grid point) has four neighbours in the grid (except boundary nodes). In terms of point coordinates in the plane: the node at $(x,y) \in [a,b] \times [c,d]$ has four neighbours that have the coordinates $(x \pm 1, y)$ or $(x, y \pm 1)$. This is the so-called *4-connected neighbourhood*, as shown in Figure 3.

    Following the threshold-free approach in TDA, the evolution of connected regions in a *superlevel set* is monitored at every value $t$ of function $f$. The superlevel set is a set of grid points in the domain of function $f$ with scalar value greater than or

10  equal to $t$. It is possible to mathematically express the superlevel set as follows

$$f^{-1}[t, +\infty) = \{(x,y) \in [a,b] \times [c,d] : f(x,y) \geq t\}. \tag{2}$$

    As $t$ is decreased connected regions of $f^{-1}[t, +\infty)$ start to appear and grow and eventually merge into larger components. Suppose there are three connected regions $(C_0, C_1, C_2)$ at value $t_0$ in a superlevel set (defined in Equation (2)), as shown in



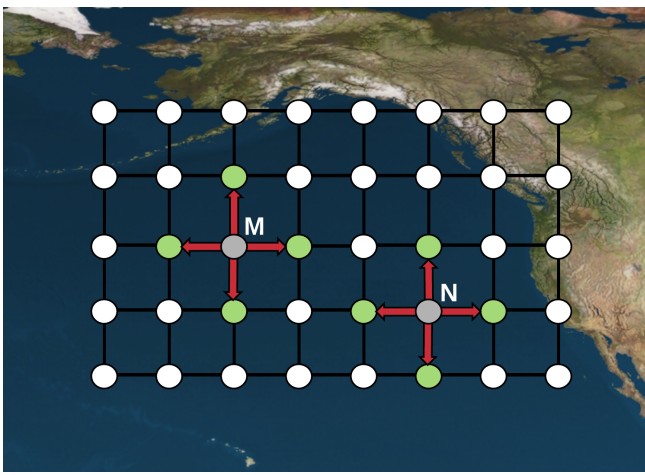

**Figure 3.** An illustration of 4-connected neighbourhood that is defined in the latitude-longitude grid in the plane with real coordinates. For example, each of the nodes $M$ and $N$ (gray points) has four neighbours, *i.e.* two nodes along the horizontal direction and two nodes along vertical direction (red arrows indicate neighbouring nodes, *i.e.* green points).

Figure 4. As values of $f$ decrease, the component $C_0$ grows until eventually, at $t_1$, it merges into the component of $C_1$, which in turn, merges into the component of $C_2$ at $t_2$, and so on.

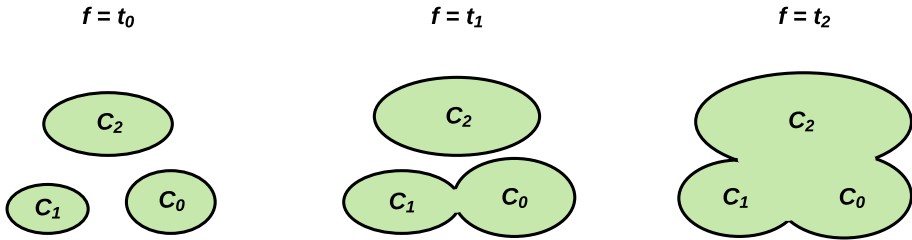

**Figure 4.** An illustration of the connected regions in the superlevel set (defined in Equation (2)) that are split into three pieces at value $t_0$. They grow and merge first at value $t_1$ and then at $t_2$ when values of function $f$ are systematically decreasing.

The above discussed approach of connected regions can be achieved by the TDA algorithm based on Union-Find data structure (U-F) (Hopcroft and Ullman, 1973). The algorithm determines connected regions of the grid by operating on sorted nodes by scalar values in decreasing order. The U-F data structure maintains the connected regions and keeps track of the evolution of these regions in the grid.

There are five main operations used in our TDA algorithm: (i) form a new connected region and add the region to the data structure; (ii) assign the right connected region to a given grid-point; (iii) check if the connected regions intersect a specified geographical location on the grid, *e.g.* we examine connected regions that intersect the west coast of North America and the latitude of the Hawaiian Islands, as shown in Figure 5 (left panel); (iv) merge two regions containing at least one same node





into one new connected region, as shown in Figure 5 (right panel); (v) track the evolution of a connected region (number of grid-points in it) as *IVW* is varied.

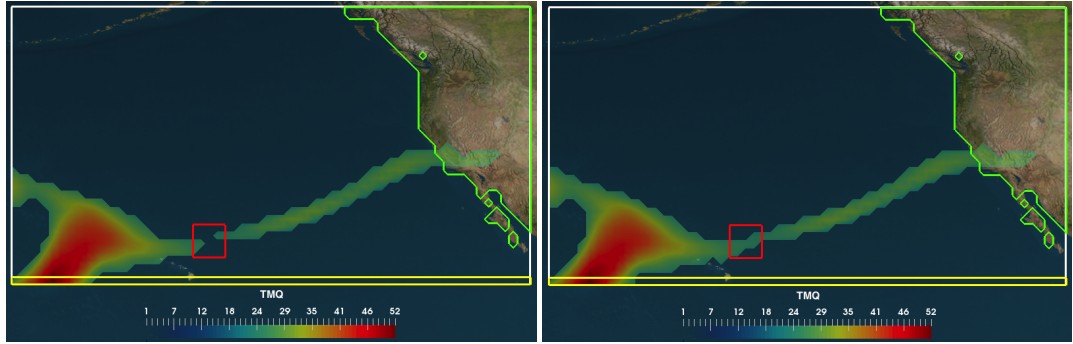

**Figure 5.** An illustration of finding connected AR regions over a specified search sector. In this example, the search for ARs is bounded by the latitude of the Hawaiian Islands (yellow line) and the west coast of North America (green line). **Left:** The red box indicates location of two regions that are disconnected at some value $IWV = t_1$. **Right:** At a new value $IWV = t_2$, where $t_2 < t_1$, the two connected regions merge into one new connected region forming a valid AR pattern. The $IVW$ (kg m$^{-2}$) plotted in this example is from the CAM5.1 climate model.

The extracted information of evolution of connected regions is encoded in *evolution plots*. The plots show the recorded number of grid points in the region as values of *IWV* are systematically decreasing, as in Figure 6. The horizontal axis $t$

contains values of *IWV* in kg m$^{-2}$ and the vertical axis $g(t)$ shows number of grid points in the connected region. The vectors from these evolution plots are encoded as a matrix with $n$ rows and $k$ columns, where $n$ is the number of time steps (snapshots) and $k$ is the size of the topological feature descriptors returned by the TDA algorithm, as is shown in the Figure 6. This $n \times k$ matrix serves as the input data to the Support Vector Machine classifier, described in the next section.

### 2.2.2 Stage 2: Applying Support Vector Machine (SVM) for Classifying Weather Patterns

Support Vector Machine is a widely used machine learning method for binary classification (recognition) task (Cortes and Vapnik, 1995; Chang and Lin, 2011). The main objective of SVM classifier is to decide whether a particular weather pattern, an AR, is present or not in a given snapshot extracted from global image. The SVM constructs a model based on the labeled topological feature descriptors in the training set and then use it to predict the labels of the testing set of the descriptors. In general, SVM finds the optimal hyperplane that separates two groups of patterns (ARs: 1 and Non-ARs: 0) by maximizing the

margin between the separating boundary and the training points closest to it (support vector).

Assume a training set of instance-labels pairs $(x_i, y_i)$, $i = 1, ..., N$, where $x_i \in \mathbb{R}^n$ and $y_i \in \{1, 0\}$. The solution of the optimization problem (finding the optimal hyperplane) is given by

$$\min_{w,b,\xi}(\frac{1}{2}w^T w + C \sum_{i=1}^{l} \xi_i), \tag{3}$$



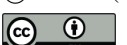

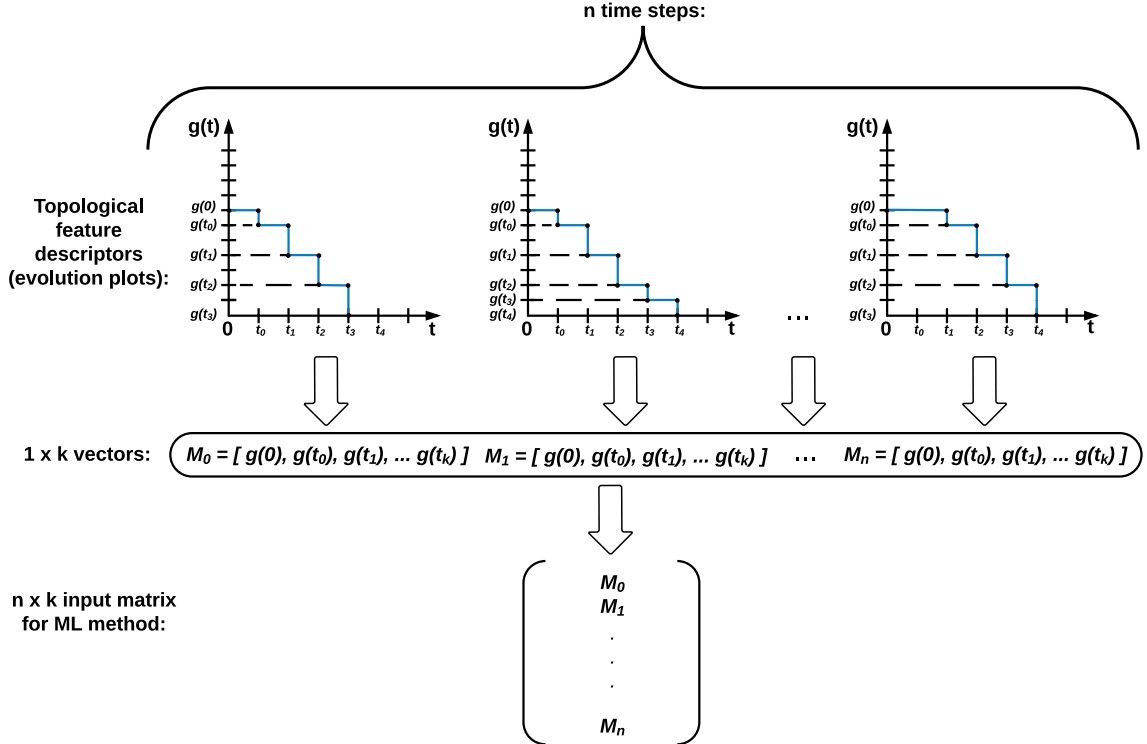

**Figure 6.** Creating an input matrix for the machine learning method: the mapped evolution plots into $1 \times k$ vectors (topological feature descriptors) are stacked on top of each other to construct $n \times k$ input matrix for Support Vector Machine classifier (SVM). The matrix is used by the SVM (see Subsection 2.2.2) along with labels provided by TECA (see Subsection 2.1) (Prabhat et al., 2015).

subject to

$$y_i(w^T\phi(x_i)+b) \geq 1 - \xi_i \ \text{ and } \ \xi_i \geq 0. \tag{4}$$

The penalty parameter of the error term takes only values greater than zero $(C > 0)$ and $\xi_i \geq 0$ is a minimum error when two groups are not linearly separable (*e.g.*, due to noise in training data). The samples $x_i$ from training set are mapped into a higher dimensional space by the kernel function to make the samples of two groups (ARs and Non-ARs) separable, as shown in Figure 7.

The kernel function that maps the input space into a higher dimensional space can be represented as $K(x_i, x_j) \equiv \phi(x_i)^T\phi(x_j)$. For this study a radial basis function (RBF) is chosen as it has been shown to achieve the best results in many applications. RBF is defined as follows

$$K(x_i, x_j) = exp(-\gamma \lVert x_i - x_j \rVert^2), \ \ \gamma > 0, \tag{5}$$

where $\gamma$ is the inverse of the standard deviation of the RBF kernel. The optimal configuration of parameters $(C, \gamma)$ is found in the experiments by applying loose grid-search and fine grid-search for these two parameters (Hsu et al., 2003).





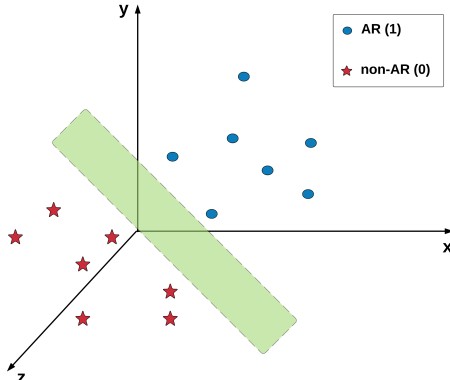

**Figure 7.** An example of a two-class dataset that is separable in some high-dimensional feature space. The SVM classifier finds a transformation of the input matrix into a high-dimensional feature space, such that in this space there exists a suitable hyperplane (green surface in the figure) that cleanly separates the data into two groups, positive (if it contains an AR) and negative (if it does not contain an AR)

## 2.3 Evaluation Metrics and Preprocessing of Data

In this subsection we define the evaluation metrics that we use to assess the reliability of our AR pattern recognition method: classification accuracy score, confusion matrix, precision score and sensitivity score. Also, we explain the preprocessing step of the input to the Support Vector Machine classifier (SVM) to address the issues of imbalanced data (He and Garcia, 2009) and data normalization (standardization).

**Classification Accuracy Score**

*Classification accuracy score* is the ratio of correct predictions of ARs to total predictions made by the machine learning classifier (in percent). *Training accuracy* is the classification accuracy obtained by applying the classifier on the training data, while *testing accuracy* is the classification accuracy for the testing data. We present the classification accuracy scores for our method in Subsection 3.2.

**Confusion Matrix**

A *confusion matrix* is a clear way to present the classification results of ARs with regard to testing accuracy of the machine learning classifier. The matrix has two rows and two columns, as shown in Table 2. The confusion matrices are shown in Subsection 3.3 and Appendix B for the SVM classifier.



**Table 2.** A confusion matrix (error matrix) is a way to present the performance of the method (typically, testing accuracy). The matrix reports the number of: (i) *False positives* - cases when the model indicates that an AR exists, when it does not in the ground truth; (ii) *False negatives* - cases when the model indicates that an AR does not exist, while in fact it does in the ground truth; (iii) *True positives* - cases when the model indicates that an AR exists, when it does in the ground truth; (iv) *True negatives* - cases when the model indicates that an AR does not exist, when it does not in the ground truth.

|  | **Label non-AR** | **Label AR** |
|---|---|---|
| **Predicted non-AR** | True negatives | False positives |
| **Predicted AR** | False negatives | True positives |

**Precision Score**

*Precision score* is a measure of the classifier's repeatability or reproducibility of ARs, and can be computed using a confusion matrix. The score is the ratio of *True positives* to the sum of *True positives* and *False positives*. It is shown in Table 7 for the SVM classifier.

**Sensitivity Score**

*Sensitivity score* is a the proportion of actual ARs which are correctly identified as ARs by the classifier. The score is the ratio of *True positives* to the sum of *True positives* and *False negatives*. It is shown in Table 7 for the SVM classifier.

**Normalizing and Balancing the Data**

*Data normalization (standardization)* is a way of adjusting measured values to a common scale (*i.e.*, $[0, 1]$) by dividing through

the largest maximum value in each feature (column of the matrix). This standardization allows for the comparison of corresponding normalized topological feature descriptors of different datasets. Also, standardization is a common requirement for many machine learning classifiers to avoid influence of outliers in training process.

*Balancing the data* is motivated by the imbalanced class problem, which is that each class of event (ARs and non-ARs) is not equally represented in the dataset. This poses a problem because SVMs tend to overfit to the majority class. We circumvent

this problem by resampling (Lemaître et al., 2017). Resampling has been applied to all matrices created by the topological data analysis algorithm along with TECA labels.

## 3   Results and Discussion

This section presents results from applying the proposed AR recognition method on test datasets. The tests have been done on CAM5.1 simulation output and MERRA-2 reanalysis product. A summary of the data, and its spatial and temporal resolution

is in Table 1. First, we compare the topological feature descriptors of ARs and non-ARs based on the ground truth labeling



provided by TECA (see Subsection 2.1). The descriptors have been normalized (see Subsection 2.3) to make the comparison of results to different datasets feasible. Second, we demonstrate performance and reliability of our method in the context of classification accuracy score obtained by the Support Vector Machine classifier. Finally, we discuss some limitations of the method, its typical failure modes (using the confusion matrix), and its precision and sensitivity in recognizing ARs.

## 3.1 Topological Feature Descriptors Representation

Topological data analysis (TDA) provides a unique way of characterizing weather events in a dataset. Figure 8 shows an example of an evolution plot with two curves of averaged topological feature descriptors. The green and magenta curves correspond to ARs and non-ARs based on the TECA labels, respectively. Each curve represents the number of grid points in the connected region measured by the TDA algorithm. Note that the TDA algorithm records the evolution of the connected region as a function of the scalar variable (here, TMQ). We observe that these two curves are close to each other, hence visually distinguishing these two groups of climate events is a challenging task. However, one can train a machine learning model, such as a Support Vector Machine (SVM), to perform this task with high accuracy.

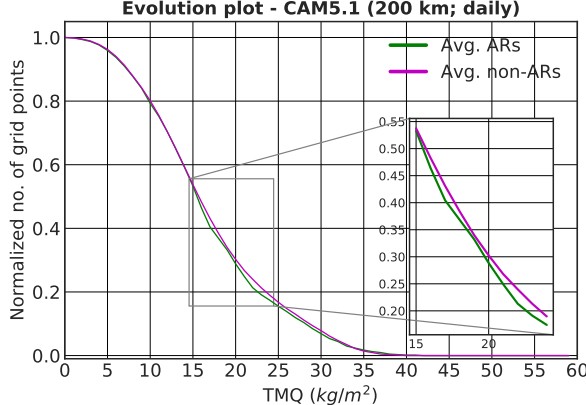

**Figure 8.** An example of normalized plots of averaged topological feature descriptors for 200 km spatial resolution and daily temporal resolution of the CAM5.1 simulation data. Note that the averaged plots for the ARs and non-ARs are very similar and it is hard to differentiate them by eye. However, a ML model can be trained to distinguish these two categories of events, by transforming the data into a high dimensional space where a unique hyperplane exists that cleanly separates the two event categories (see Subsection 7).

Figures 9 and 10 show the evolution plots of topological feature descriptors obtained for both AR and non-AR events. Each plot consists of 100 randomly chosen examples of both categories of events from the raw 2D snapshots from both the CAM5.1 climate model and the MERRA-2 reanalysis product. As in Figure 8, we observe that it is hard to differentiate by eye the topological feature descriptor curves for ARs versus non-ARs. Yet the trained SVM can distinguish between AR and non-AR with fairly high accuracy by learning a suitable transformation of the feature descriptor curves into some high dimensional space, where there exists a clean separation of the AR and non-AR groups with a suitable hyperplane (as shown in Figure 7).





This is typical in image recognition tasks, i.e. features that are difficult to distinguish by the human eye can be learned by a suitable ML method in order to perform the classification task with high accuracy.

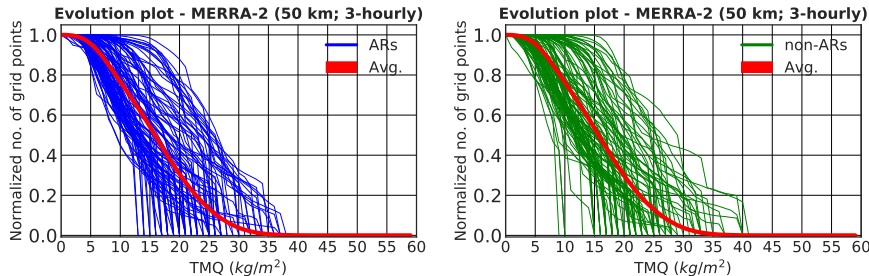

**Figure 9.** Normalized evolution plots of averaged (red curves) and 100 arbitrarily selected topological feature descriptors of ARs (blue curves; upper row) and non-ARs (green curves; lower row). For 3-hourly temporal resolution and 50 km spatial resolution of the MERRA-2 reanalysis data. The plots illustrate how topological descriptors vary versus *IWV*. They show the topological representation of both AR and non-AR events for MERRA-2 reanalysis data.

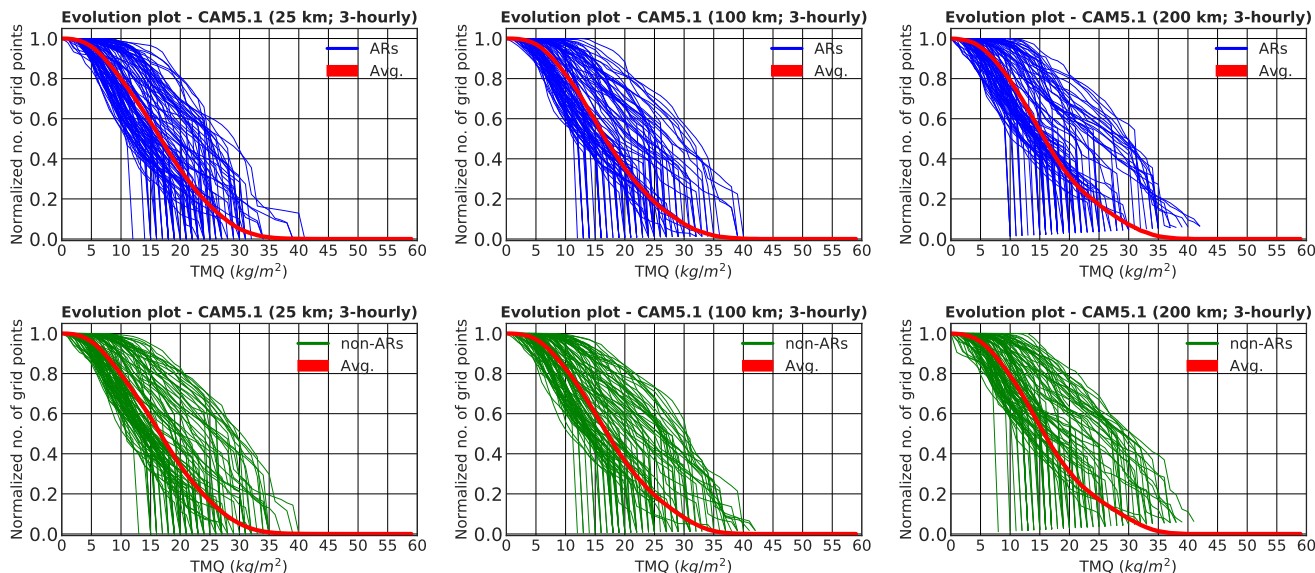

**Figure 10.** Normalized evolution plots of averaged (red curves) and 100 arbitrarily selected topological feature descriptors of ARs (blue curves; upper row) and non-ARs (green curves; lower row). For 3-hourly temporal resolution and 25 km (left), 100 km (middle), and 200 km (right) spatial resolutions of the CAM5.1 simulation data. The plots illustrate how topological descriptors vary versus *IWV*. They show the topological representation of both AR and non-AR events for each resolution of CAM5.1 model output.




The same analyses using topological feature descriptors has been done for all other datasets listed in Table 1, *i.e.* similar evolution plots have been prepared for daily temporal resolution and three different spatial resolutions of CAM5.1 model. We note that the curves look similar, hence we only show one set of cases, and the others can be found in Appendix A.

## 3.2 Classifier Performance

5 We now evaluate the performance and reliability of the proposed AR recognition method by measuring the classification accuracy (as defined in Section 2.3). Tables 3, 4 and 5 summarize the classification accuracy of our method for the CAM5.1 climate model at different horizontal resolutions as well as for the MERRA-2 reanalysis product. Training accuracy measures how well the model learns from training data, *i.e.* "ground truth" data labeled with ARs and non-ARs. Testing accuracy measures how well the method performs on a "held out" dataset.

**Table 3.** Classification accuracy score estimated of SVM classifier for 3-hourly temporal resolution of CAM5.1 model with three different spatial resolutions. Table also show number of snapshots (# of events for both categories: ARs and non-ARs).

| Dataset | Training Accuracy | Testing Accuracy | # of AR snapshots | # of Non-AR snapshots |
|---|---|---|---|---|
| CAM5.1 (25 km) | 83% | 83% | 6838 | 6848 |
| CAM5.1 (100 km) | 77% | 77% | 7182 | 7581 |
| CAM5.1 (200 km) | 90% | 90% | 3914 | 3914 |

10 Table 3 shows that the SVM classifier is able to learn to differentiate better ARs from non-ARs when the spatial resolution of the climate model is lower. We speculate that the reason for this is that despite the fact that the higher resolution version of the model more realistically represents AR statistics (Wehner et al., 2014), the *IWV* fields tend to be noisier, leading to a less smooth topological representation and lower training accuracy. Further, despite a lower number of ARs to train on or classify due to resolution effects, the fairly high testing classification accuracy for the CAM5.1 (200 km) suggests that the SVM is able 15 to capture key nonlinear dependencies between topological feature descriptors.

**Table 4.** Classification accuracy score estimated of SVM classifier for daily temporal resolution of CAM5.1 model with three different spatial resolutions. Table also show number of snapshots (# of events for both categories: ARs and non-ARs).

| Dataset | Training Accuracy | Testing Accuracy | # of AR snapshots | # of Non-AR snapshots |
|---|---|---|---|---|
| CAM5.1 (25 km) | 78% | 82% | 624 | 624 |
| CAM5.1 (100 km) | 85% | 84% | 700 | 700 |
| CAM5.1 (200 km) | 89% | 91% | 397 | 397 |





In Table 4 we observe a similar trend with classification accuracy and model resolution as in Table 3. Also note that the number of snapshots is about 10 times smaller, but this does not affect testing accuracies (consistently above 80%). This suggests that even though event boundaries may be more smeared out in daily averages, the topological feature descriptors encode sufficiently unique information about ARs and non-ARs that SVM is able to distinguish between the two categories

with high accuracy. Finally, the SVM has highest training and testing accuracy for CAM5.1 (200 km), as in Table 3.

**Table 5.** Classification accuracy score of SVM classifier for 3-hourly temporal resolution and 50 km spatial resolution of MERRA-2 reanalysis. Table also shows number of snapshots (# of events for both categories: ARs and non-ARs).

| Dataset | Training Accuracy | Testing Accuracy | # of AR snapshots | # of Non-AR snapshots |
|---------|-------------------|------------------|-------------------|------------------------|
| MERRA-2 (50 km) | 80% | 80% | 13294 | 13434 |

Table 5 reports the classification accuracy of SVM for MERRA-2 reanalysis product. Note that classification accuracies are about the same as for of 3-hourly datasets from the CAM5.1 model. Hence we conclude that the SVM classification method is robust to the source of maps of *IWV*.

In summary, the model has consistently high classification accuracy for ARs (77% - 91%) across a broad set of spatial

and temporal resolutions, illustrating that the combination of topological data analysis and machine learning is an effective and efficient threshold-free strategy for detecting ARs in large climate datasets. We note that the ML model is biased by the "ground truth" data produced by TECA using the threshold based criteria for ARs identification. Characterizing the influence of using different ground truth data is beyond the scope of this study.

### 3.3   Limitations of our method

In this section we examine some limitations of the proposed method. We investigate some typical failure modes further by examining snapshots of mis-classified events. Then we use the confusion matrix, and precision and sensitivity scores to quantify how accurately and precisely the model is able to classify events by comparing against ground truth data.

Figure 11 shows a typical failure mode of the proposed method: examples of AR misclassified as non-ARs, *i.e. false negatives*. We note that imperfect training data is a challenge in ML and high quality ground truth data is essential for good model

performance. However, in some cases, the process of feature abstraction that occurs while training the ML model may indeed produce a model that could outperform the algorithms used for producing the original "ground truth" training data. Figure 12 shows another typical failure mode of the proposed method: examples of non-ARs misclassified as ARs, *i.e. false positives*. This failure mode is often related to the merging of multiple events, either of two ARs or of an AR (left panel) and some other event with high concentration of water vapor and similar topological structure, such as an extra-tropical cyclone (ETC)

(right panel). We use the confusion matrix (described in Section 2.3) to give more insight into the classification accuracy of the method and to quantitatively compare the types of correct and incorrect predictions made, as shown in Table 6 for CAM5.1 model output at 25km spatial resolution and 3-hourly temporal resolution. Note that the model performs very well in clas-

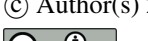



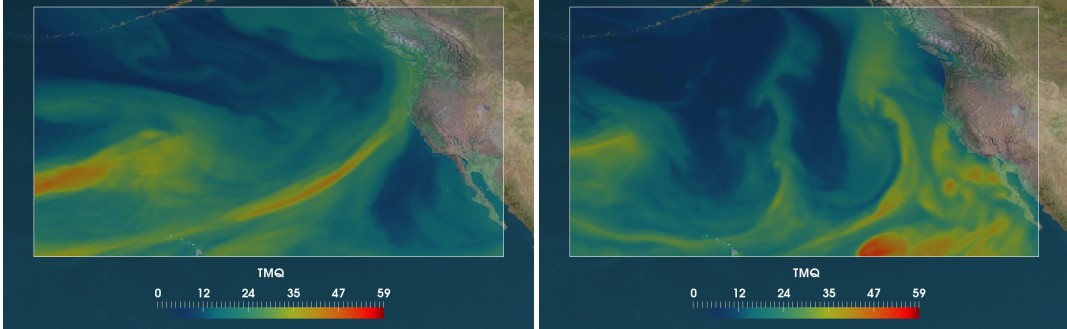

**Figure 11.** Sample images of events from the testing set showing a typical failure mode of the proposed method: examples of ARs misclassified as non-ARs (false negatives). Figure shows *IWV* ($\mathrm{kg\,m}^{-2}$) in the CAM5.1 climate model (color map) and the land-sea-mass as background (satellite image). **Left:** The model fails likely because there are two separate events in the figure, one is a fully formed AR and another is the start of a new AR; **Right:** The method fails likely due to imperfect training data. The "ground truth" data from TECA labels this image as an AR, although (visually) it does not appear to satisfy the definition of an AR. This illustrates how imperfect training data, due to limitations of the algorithm used to produce ground truth data, impacts the performance of the ML model.

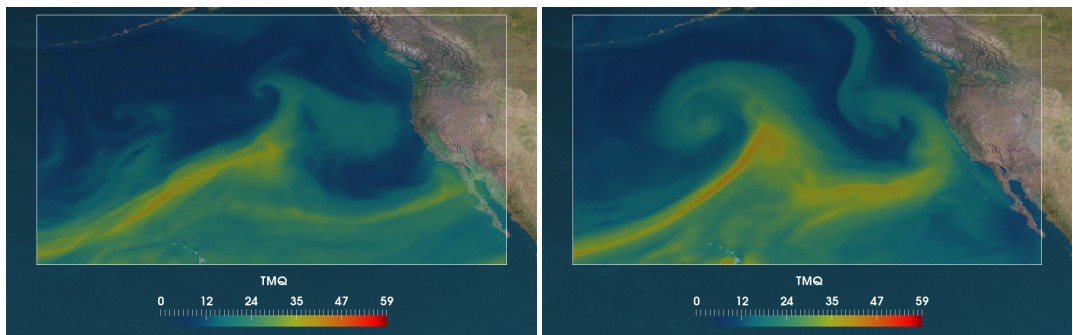

**Figure 12.** Sample images of events from the testing set showing a typical failure mode of the proposed method: examples of non-ARs misclassified as ARs (false positives). Figure shows *IWV* ($\mathrm{kg\,m}^{-2}$) in the CAM5.1 climate model (color map) and the land-sea-mass as background (satellite image). **Left:** The model likely fails because of the presence of two AR-like branches that are close to each other, one that has not yet made landfall and another that probably remains after previous event; **Right:** The model fails likely due to the merging of two events, both with high concentration of water vapour, one that appears to be an AR and the other likely an extra-tropical cyclone (ETC).

sifying AR events correctly but has relatively poorer performance for non-AR events. In Appendix B we present confusion matrices of the method for different spatial and temporal resolutions of the CAM5.1 model and MERRA-2 reanalysis product.

Table 7 shows that the method has the highest precision and sensitivity scores for $200\,\mathrm{km}$ resolution of CAM5.1 model for both 3-hourly and daily temporal resolutions. The scores are slightly lower for other spatial and temporal resolutions of
5  CAM5.1 and reanalysis data.



**Table 6.** Confusion matrix of the method for testing set - the CAM5.1 data (3-hourly, 25 km), which shows the numbers of correctly classified (diagonal) and incorrectly classified events (off-diagonal).

|                    | Label non-AR | Label AR |
| ------------------ | ------------ | -------- |
| **Predicted non-AR** | 5047         | 391      |
| **Predicted AR**     | 1432         | 4078     |

**Table 7.** Precision and sensitivity scores (described in section 2.3) calculated for all datasets listed in Table 1. Both scores show the ability of the SVM classifier in assigning correct labels to the test instances.

| Dataset | Precision | Sensitivity |
| --- | --- | --- |
| **CAM5.1 (25km, 3-hourly)** | 0.91 | 0.74 |
| **CAM5.1 (100km, 3-hourly)** | 0.83 | 0.67 |
| **CAM5.1 (200km, 3-hourly)** | 0.95 | 0.85 |
| **CAM5.1 (25km, daily)** | 0.87 | 0.77 |
| **CAM5.1 (100km, daily)** | 0.86 | 0.83 |
| **CAM5.1 (200km, daily)** | 0.97 | 0.85 |
| **MERRA-2 (25km, 3-hourly)** | 0.84 | 0.74 |

## 4 Conclusions and Future Work

In this paper, we propose a novel and automated method for recognizing AR patterns in large climate datasets. The method combines topological data analysis (TDA) with machine learning (ML), both of which are powerful tools that the climate science community often does not use.

We show that the proposed method is reliable, robust and performs well by testing it on a wide range of spatial and temporal resolutions of CAM5.1 climate model output as well as the MERRA-2 reanalysis product. The "ground truth" labels are obtained using TECA (Prabhat et al., 2015). The performance of the method is quantified by its classification accuracy in recognizing AR events, and precision and sensitivity scores.

Despite background noise, low intensity AR signals and the existence of other events within the 2D snapshots, our method
is shown to work well. The method tends to perform better for lower resolution data and we speculate that this is because high resolution simulations tend to produce noisier spatial patterns, which tend to confuse the machine learning model more easily than low resolution simulations.

The key advantage of the topological feature descriptors used in this work is that it is a threshold-free method that succinctly encapsulates the most important topological features of ARs. We anticipate that because the method is threshold-free, it can be
applied to different climate change scenarios without any tuning, which we will be exploring in future work.



Further, it is a much faster method than, for example, using convolutional neural networks (Liu et al., 2016) (processing time of a couple of minutes versus a few days).

In future work, we will test our method on direct observations via satellite images. We propose that the method can be made more robust by (i) employing a full "persistence" concept from TDA; and (ii) training SVM on ground truth data that are not biased by fixed threshold criteria. This study shows that the TDA and ML framework could be an effective way to characterize and identify a wide range of other weather and climate phenomena, such as blocking events and jet streams. As the TDA step is not restricted to a 2D scalar field on a grid, it is also possible to apply to higher-dimensional or multivariate fields. A similar TDA-based approach has successfully been applied to data skeletonization (Kurlin, 2015) and segmentation (Kurlin, 2016) problems. Hence, we believe that this method can be extended to be applied in a variety of other climate science problems where defining suitable thresholds remains a challenge.





## Appendix A: Additional evolution plots for daily temporal resolution of CAM5.1 climate model output

This appendix contains additional evolution plots mentioned in Subsection 3.1.

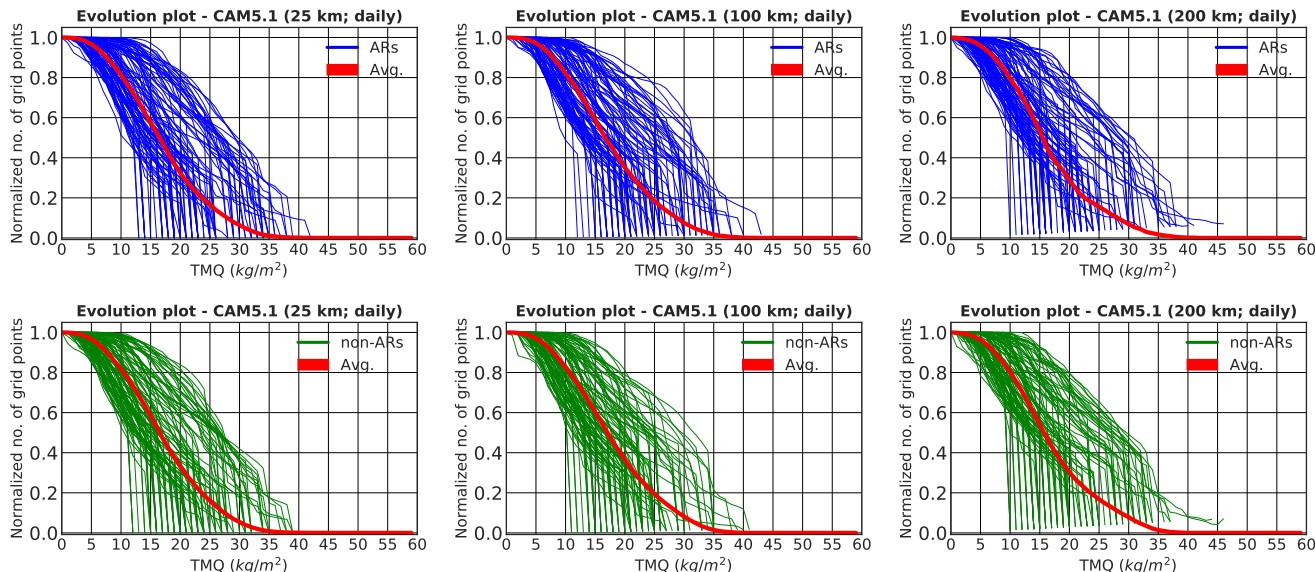

**Figure A1.** Normalized evolution plots of averaged (red curves) and 100 arbitrarily selected topological feature descriptors of ARs (blue curves; upper row) and non-ARs (green curves; lower row). For daily temporal resolution and 25 km (left), 100 km (middle), and 200 km (right) spatial resolutions of the CAM5.1 simulation data. The plots illustrate the relationship of topological descriptors variation versus average. They show the topological representation of both AR and non-AR events for each resolution of CAM5.1 model output.





**Appendix B: Additional confusion matrices for CAM5.1 and MERRA-2 testing sets**

This appendix includes the rest of the confusion matrices (tables) that were considered in Subsection 3.3. The presented tables allow for quantitative comparison of ML classifier performance to recognize ARs in CAM5.1 climate model outputs and MERRA-2 reanalysis product.

**Table B1.** Confusion matrix of the method on testing set - the MERRA-2 data (3-hourly, 50 km). It shows the numbers of correctly recognized (the diagonal) and the number of incorrectly classified events (off-diagonal).

|                     | Label non-AR | Label AR |
| ------------------- | :----------: | :------: |
| **Predicted non-AR** | 9211         | 1489     |
| **Predicted AR**     | 2782         | 7900     |

**Table B2.** Confusion matrix of the method for testing set - the CAM5.1 data (3-hourly, 100 km). It shows the numbers of correctly recognized (the diagonal) and the number of incorrectly classified events (off-diagonal).

|                     | Label non-AR | Label AR |
| ------------------- | :----------: | :------: |
| **Predicted non-AR** | 5258         | 808      |
| **Predicted AR**     | 1887         | 3857     |

**Table B3.** Confusion matrix of the method for testing set - the CAM5.1 data (3-hourly, 200 km). It shows the numbers of correctly recognized (the diagonal) and the number of incorrectly classified events (off-diagonal).

|                     | Label non-AR | Label AR |
| ------------------- | :----------: | :------: |
| **Predicted non-AR** | 3020         | 137      |
| **Predicted AR**     | 466          | 2639     |





**Table B4.** Confusion matrix of the method on testing set - the CAM5.1 data (daily, 25 km). It shows the numbers of correctly recognized (the diagonal) and the number of incorrectly classified events (off-diagonal).

|  | Label non-AR | Label AR |
|---|---|---|
| **Predicted non-AR** | 444 | 59 |
| **Predicted AR** | 116 | 379 |

**Table B5.** Confusion matrix of the method for testing set - the CAM5.1 data (daily, 100 km). It shows the numbers of correctly recognized (the diagonal) and the number of incorrectly classified events (off-diagonal).

|  | Label non-AR | Label AR |
|---|---|---|
| **Predicted non-AR** | 486 | 77 |
| **Predicted AR** | 97 | 460 |

**Table B6.** Confusion matrix of the method for testing set - the CAM5.1 data (daily, 200 km). It shows the numbers of correctly recognized (the diagonal) and the number of incorrectly classified events (off-diagonal).

|  | Label non-AR | Label AR |
|---|---|---|
| **Predicted non-AR** | 306 | 8 |
| **Predicted AR** | 48 | 273 |

*Code and data availability.* The data and source code are available at NERSC Science Gateway: https://doi.org/10.25342/GMD_2018.

*Competing interests.* The authors declare that they have no conflict of interest.

*Acknowledgements.* Grzegorz Muszynski and Vitaliy Kurlin would like to acknowledge Intel for supporting the IPCC at University of Liverpool. Karthik Kashinath was supported by the Intel Big Data Center, and Michael Wehner was supported by the Regional and Global

5 Climate Modeling Program of the Office of Biological and Environmental Research in the Department of Energy Office of Science under contract number DE-AC02-05CH11231. This research used resources of the National Energy Research Scientific Computing Center, a DOE Office of Science User Facility supported by the Office of Science of the U.S. Department of Energy under Contract No. DE-AC02-05CH11231.



   This document was prepared as an account of work partially sponsored by the United States Government. While this document is believed
to contain correct information, neither the United States Government nor any agency thereof, nor the Regents of the University of California,
nor any of their employees, makes any warranty, express or implied, or assumes any legal responsibility for the accuracy, completeness, or
usefulness of any information, apparatus, product, or process disclosed, or represents that its use would not infringe privately owned rights.

5  Reference herein to any specific commercial product, process, or service by its trade name, trademark, manufacturer, or otherwise, does not
necessarily constitute or imply its endorsement, recommendation, or favoring by the United States Government or any agency thereof, or the
Regents of the University of California. The views and opinions of authors expressed herein do not necessarily state or reflect those of the
United States Government or any agency thereof or the Regents of the University of California.





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
