# Peer review of "Topological Data Analysis and Machine Learning for Recognizing Atmospheric River Patterns in Large Climate Datasets"

_Geoscientific Model Development, 2018_

## Referee Comment (RC1) · Anonymous Referee #1 · 28 Jun 2018

General Comment

This paper presents a method to detect atmospheric rivers by using topological data analysis and machine learning techniques. The method is novel and different with most of methods to detect atmospheric river (AR) in literature, as well as has the advantage of not depending on sometime arbitrary values as threshold for isolating potential ARs. In my opinion, this paper deserves to be published, however, I provide a few comments below which should be easily addressed by the authors in order to improve the manuscript.

Specific comments:

1. Since the method is so novel and different with respect to relatively large number of method available now in literature (e.g., Shields et al 2018), it would be nice to see some figures and analysis of the frequency of ARs that make landfall on the west coast of North America according to this method. These extra figures may constitute a reference point to compare the occurrence of AR between this new and novel method and previous ones in other studies, especially those that use IWV field to detect ARs (Neiman et al 2008, Dettinger et al 2011, Wick et al 2013). A comparison with previous climatology is expected, as well as between the different gridded datasets used here and the possible cause of differences.

2. It is not clear for me what labels are uses as "ground truth". It should be stated explicitly in the text despite that a related reference has been added.

3. It is suggested in page 5 line 20-23 that the TDA approach works with scalar field. If so, this is the main reason to not use the vectoral IVT variable for identifying ARs, rather than IWV being observable by satellite. Please explain further about this point, especially because IVT variable is now being used more than IWV variable for detecting ARs (e.g., see ARTMIP paper, Shields et al 2018).

---

## Referee Comment (RC2) · S. Thao (Referee) · 9 Oct 2018

General comments:

The paper presents a method to detect atmospheric rivers (ARs) in climate datasets. Unlike most existing methods, this one relies on marching learning and learns a classification rule for the detection of ARs based on a training dataset. In my opinion, one novelty of the paper lies in the choice of the features used for the classification. From maps of integrated water vapor, new features are constructed from topological data analysis that could me more suited for the problem.

In general, I think that the paper is well written and I appreciate the pedagogical effort made to clearly explain the methodology as well as the illustrations of cases where the algorithm performs well and not so well. Hence, I don't see any major reasons not to published the paper. I only have a few comments and suggestions that I think could benefit the paper.

Specific comments:

1. I find the use of the term "threshold-free" is maybe not the most appropriate. While I understand that in most of the cases, "threshold-free" means that the method does not rely on a fixed, predetermined, arbitrary threshold for the detection of ARs, thresholds are still used several times during the proposed procedure. Indeed, the goal of the SVM step is still to learn a threshold to separate the ARs from non-ARs from the training set and the topological features. The topological features are also constructed from a set of thresholds. (And to be more provocative, for now, the labels in the training set were also generated by an AR detection methods using thresholds). For me, the value of the paper is that it shows that if a we have a good training dataset, there is more efficient way to build this decision threshold than manually tinkering parameters of the classifier/detector.

2. In the same way, I am not sure I understand the following sentence from the abstract and the conclusion (p17, l-14-15) : "We anticipate that because the method is threshold-free, it can be 15 applied to different climate change scenarios without any tuning". If the statistical relationships between features and the target variable change through time, should not you retrain the SVM as the other methods have to reevaluate their thresholds ?

3. I think the explanation on the SVM could be improved if Figure 7 was split into 2: the first figure would illustrate the (linear) SVM and the different quantities in equations (3) and (4) (see e.g. https://en.wikipedia.org/wiki/Support_vector_machine#/media/File:Svm_max_sep_hyperplane_with_margin.png).

The second figure would focus more on the "kernel trick". For instance, it would show a case were a linear classifier could not separate the two classes in a 2D space but would managed to do it if data were mapped into a 3D space.

4. (P9, l7) "The kernel function that maps the input space into a higher dimensional space ...". I think the sentence can be a little bit nuanced. As far as I understand, the kernel function returns the inner product between two points projected into higher dimensional space by a mapping function phi. Each kernel function is implicitly associated with a mapping function phi (which does not need to be known for an actual application and that's one of the strong point of kernel methods). That's why the function phi is called a kernel induced implicit mapping.

5. (P9, l12) " applying loose grid-search and fine grid-search for these two parameters". Do you use grid search with some kind of cross-validation scheme?

6. I think it should be clearly mentioned in the main text or in a table how many data points were used in the training set and the test sets. We could try to deduce it from confusion matrices but it is not very practical.

7. In the same way, for table 3, 4, etc . . . , the number of snapshots mentioned, is it for the test or training sets ?

8. (P18, l1), Authors compare the computing time of their algorithm with the one of Liu et al. (2016) thats uses deep learning. How do both methods compare in terms of performances ?

9. For the sake of reproducibility, it would be nice to at least provide in supplementary materials, details about the actual implementation of the methods. For instance, the programming language used, the potential external softwares/packages/libraries used and for which step of the method.

---

## Author Response (AR1)

C: Referee comments are in red text with italics.

A: Author's responses are in black text.

N: New additions (major changes) to the manuscript, where applicable, are noted in green text.

**Response to GMDD Interactive comment from Reviewer #1**

**General comments**

C: This paper presents a method to detect atmospheric rivers by using topological data analysis and machine learning techniques. The method is novel and different with most of methods to detect atmospheric river (AR) in literature, as well as has the advantage of not depending on sometime arbitrary values as threshold for isolating potential ARs. In my opinion, this paper deserves to be published, however, I provide a few comments below which should be easily addressed by the authors in order to improve the manuscript.

A: We thank the Reviewer for noting the novelty and the advantage of the method presented in this paper.

**Specific comments**

C: Since the method is so novel and different with respect to relatively large number of method available now in literature (e.g., Shields et al 2018), it would be nice to see some figures and analysis of the frequency of ARs that make landfall on the west coast of North America according to this method. These extra figures may constitute a reference point to compare the occurrence of AR between this new and novel method and previous ones in other studies, especially those that use IWV field to detect ARs (Neiman et al 2008, Dettinger et al 2011, Wick et al 2013). A comparison with previous climatology is expected, as well as between the different gridded datasets used here and the possible cause of differences.

A: We thank the reviewer for this suggestion. This paper intends to describe in detail the novel method proposed and validate its usefulness in recognizing AR patterns. While it would indeed be nice to see some figures of the statistics of ARs on the west coast of North America using our novel approach, a part of this analysis has been done for MERRA-2 data for the Atmospheric River Inter-comparison Project (ARTMIP) (https://www.geosci-model-dev.net/11/2455/2018/). Further analysis of the statistics of ARs using this method is underway and in preparation for a subsequent paper with the ARTMIP. Furthermore, we plan to repeat this analysis for the CAM5.1 data also for the ARTMIP. We intend to publish our results on the frequency analyses and other relevant statistics for the next ARTMIP paper, which is in preparation.

C: It is not clear for me what labels are uses as "ground truth". It should be stated explicitly in the text despite that a related reference has been added.

A: We thank the reviewer for pointing this out. The definition of the "ground truth" has been included on page 4, lines 24-29 in the paper. The "ground truth" is a set of binary labels (AR: 1, Non-AR: 0) generated for all datasets listed in Table 1 by a heuristics-based detection method implemented in the Toolkit for Extreme Climate events Analysis (TECA) (Prabhat et al., 2015). TECA includes an AR detection method that uses geometrical constraints and a fixed threshold parameter on IWV based on threshold value provided in (Dettinger et al., 2011). We are also aware

that the ground truth data used for training comes from a method that uses thresholds, and hence there is the possibility of a bias, but this is a classic problem in machine learning and we plan to address the sensitivity to choice of ground truth in future work.

C: It is suggested in page 5 line 20-23 that the TDA approach works with scalar field. If so, this is the main reason to not use the vectoral IVT variable for identifying ARs, rather than IWV being observable by satellite. Please explain further about this point, especially because IVT variable is now being used more than IWV variable for detecting ARs (e.g., see ARTMIP paper, Shields et al 2018).

A: We agree with the reviewer that the precise reason for using IWV should be clarified. While the TDA method implemented in this novel approach uses scalar fields, the method itself is applicable to vector fields (like IVT) and point clouds. Hence a scalar field is neither a requirement nor is it a restriction of the choice of TDA method used here. We plan to incorporate an extension of this method to vector fields (IVT) and point clouds in future work. As mentioned in the paper, we wish to emphasize that the choice of using IWV is because of the intention for this method to be usable across any type of dataset, including direct observational data, where IVT is very hard to measure directly. While we have not tested the method on direct observational data, it can be done without any extensions or modifications to the current method in its current form.

**Response to GMDD Interactive comment from Reviewer #2 (Soulivanh Thao)**

**General comments**

C: The paper presents a method to detect atmospheric rivers (ARs) in climate datasets. Unlike most existing methods, this one relies on marching learning and learns a classification rule for the detection of ARs based on a training dataset. In my opinion, one novelty of the paper lies in the choice of the features used for the classification. From maps of integrated water vapor, new features are constructed from topological data analysis that could me more suited for the problem. In general, I think that the paper is well written and I appreciate the pedagogical effort made to clearly explain the methodology as well as the illustrations of cases where the algorithm performs well and not so well. Hence, I don't see any major reasons not to published the paper. I only have a few comments and suggestions that I think could benefit the paper.

A: We thank the reviewer for the valuable comments, which we also think will improve the quality of the paper.

**Specific comments**

C: I find the use of the term "threshold-free" is maybe not the most appropriate. While I understand that in most of the cases, "threshold-free" means that the method does not rely on a fixed, predetermined, arbitrary threshold for the detection of ARs, thresholds are still used several times during the proposed procedure. Indeed, the goal of the SVM step is still to learn a threshold to separate the ARs from non-ARs from the training set and the topological features. The topological features are also constructed from a set of thresholds. (And to be more provocative, for now, the labels in the training set were also generated by an AR detection methods using thresholds). For me, the value of the paper is that it shows that if a we have a good training dataset, there is more efficient way to build this decision threshold than manually tinkering parameters of the classifier/detector.

A: We thank the reviewer for pointing out the potential confusion in the term "threshold-free". To clarify, we are specifically referring to the deficiency of most existing AR-detection techniques that use a fixed, predetermined, arbitrary threshold on certain physical variables in order to detect ARs. In our approach, the topological feature extraction is threshold free in the sense that we do not choose any fixed or predetermined thresholds to calculate features for the detection of ARs. In particular, topological features (invariants), in this case connected components/regions, are computed under all possible values of certain parameter, here the Integrated Water Vapour (IWV). This implies that the connected components do not rely on specific choice of a threshold value, i.e. "threshold-free".

However, the term "threshold-free" may be confusing for readers because indeed the training data uses a heuristic algorithm that has built-in thresholds on IWV. We mention on page 15, lines 11-13 and on page 18, lines 3-5 that an imperfect "ground truth" training set generated by the AR detection heuristic implemented in TECA (Prabhat, et al., 2015) is biased by using the fixed threshold based criteria for AR identification. In future work, we plan to test the method by training the classifier on datasets that are manually labelled. This should circumvent the problem of the classification results biased by fixed thresholds used for generating the ground truth data.

On the other hand, we would like to clarify that the SVM does not learn a threshold from the topological features to separate the ARs from non-ARs. Instead, the SVM finds a transformation of the topological vectors into a high dimensional space where ARs and non-ARs are separable by a suitable hyper-plane (for clarity this has been included in the paper on page 8, lines 11-15).

C: In the same way, I am not sure I understand the following sentence from the abstract and the conclusion (p17, l-14-15) : "We anticipate that because the method is threshold-free, it can be applied to different climate change scenarios without any tuning". If the statistical relationships between features and the target variable change through time, should not you retrain the SVM as the other methods have to reevaluate their thresholds?

A: Yes, if the distribution of data and target variables change over time, the SVM model should be retrained. However, by "... it can be applied to different climate scenarios without any tuning." we refer to "Stage 1" of the method, i.e. topological feature extraction. To clarify, there is no need to determine any threshold criteria for this topology-based AR detection method. Hence, when the spatial resolution of the climate model changes or a different climate scenario is examined, there is no parameter re-tuning, unlike in the case of heuristic methods used by most other AR-detection methods, e.g., TECA (Prabhat, et al., 2015). We rephrase as follows: "We anticipate that because the method is threshold-free (there is no need to determine any threshold criteria for the TDA step), when the spatial resolution of the climate model changes, there is no parameter re-tuning, unlike in the case of heuristic methods used by most other AR-detection methods. An application of this method to different climate change scenarios without any tuning will be explored in future work.", and "An advantage of the proposed method is that it is threshold-free—there is no need to determine any threshold criteria for the detection method—when the spatial resolution of the climate model changes." These texts can be found in the revised manuscript on page 18, lines 14-17, and on page 1, lines 6-8, respectively.

C: I think the explanation on the SVM could be improved if Figure 7 was split into 2: the first figure would illustrate the (linear) SVM and the different quantities in equations (3) and (4) (see for e.g. https://en.wikipedia.org/wiki/Support_vector_machine#/media/File:Svm_max_sep_hyperplane_with_margin.png). The second figure would focus more on the "kernel trick". For instance, it would show a case were a linear classifier could not separate the two classes in a 2D space but would managed to do it if data were mapped into a 3D space.

A: We thank the reviewer for this excellent suggestion of dividing Figure 7 into two figures (i.e., Fig. 7 and Fig. 8). These two figures and captions have been attached to this response and are included in the revised manuscript on page 9.

Fig. 7: An example of linear SVM that finds the optimal hyperplane $w^T\phi(x) + b = 0$, its maximum-margin $\frac{2}{\sqrt{w^T w}}$ separating samples from two classes in data (blue dots and red stars), and all other quantities in the equations (3), (4). $\zeta$ is a variable defining how much on the 'wrong' side of the hyperplane a sample is: if it is $1 > \zeta > 0$, the point is classified correctly, but by less of a margin than the optimal hyperplane was found, else if it is more than $\zeta > 1$, the point is classified incorrectly. The magenta dot indicates an example of misclassified sample from the class of blue dots. Support vectors help to find the margin for the optimal linear hyperplane. $\phi(x)$ is a linear transformation in this case.

Fig. 8: a) An example of no clear linear separation between two classes (e.g., ARs and non-ARs) in data. This case cannot be solved using linear SVM. b) In a situation where the set of two class samples is not linearly separable in the original space the SVM introduces the notion of a 'kernel function induced feature space' which casts the data into a higher dimensional space where the data is separable.

C: (P9, l7) "The kernel function that maps the input space into a higher dimensional space ...". I think the sentence can be a little bit nuanced. As far as I understand, the kernel function returns the inner product between two points projected into higher dimensional space by a mapping function phi. Each kernel function is implicitly associated with a mapping function phi (which does not need to be known for an actual application and that's one of the strong point of kernel methods). That's why the function phi is called a kernel induced implicit mapping.

A: Yes, this sentence should be rewritten to avoid confusion. We rephrase as follows: "The samples $\{x_i\}$, where $x_i \in R^n$, from the training set are mapped into a high dimensional feature space $F$ by means of the transformation $\phi(x_i)$, where $\phi(x) : R^n \to F$. This transformation makes the samples of two groups (ARs and Non-ARs) separable, as shown in Figure 8. Then, the similarity between observations $x_i$ and $x_j$ is computed by kernel function $K(x_i, x_j)$ that can be expressed as an inner product $\langle\phi(x_i), \phi(x_j)\rangle_F$ in the feature space $F$. Hence, it is sufficient to know $K(x_i, x_j) = \langle\phi(x_i), \phi(x_j)\rangle_F$ rather than $\phi(x)$ explicitly (Burges, 1998)." This text can be found in the revised manuscript on page 9, lines 4-8.

C: (P9, l12) "applying loose grid-search and fine grid-search for these two parameters". Do you use grid search with some kind of cross-validation scheme?

A: Yes, we used a stratified k-fold cross-validated grid-search to find optimal values of parameters $C$ and $\gamma$ regarding the SVM classification performance (Hsu et al., 2003). We split a training set into $k$ folds of equal size in a such manner that $k$ folds do not overlap one another. Then, iteratively, one fold was chosen for testing and the remaining k-1 folds were used for training the classifier. Since a grid-search is time-consuming we used a coarse grid-search on training sets to identify the range of $C$ and $\gamma$ values with respect to classification performance. Then, we conducted a fine grid-search for the identified range of the parameters to select the optimal parameters regarding classification performance.

C: I think it should be clearly mentioned in the main text or in a table how many data points were used in the training set and the test sets. We could try to deduce it from confusion matrices but it is not very practical.

A: This comment is included in the revised manuscript on page 14, lines 8-9.

C: In the same way, for table 3, 4, etc . . . , the number of snapshots mentioned, is it for the test or training sets?

A: The number of snapshots mentioned in the manuscript represents the total number of samples of both classes (AR & Non-AR) after resampling was applied to the original datasets due to the imbalanced class problem. Resampling to solve the class imbalance problem has been mentioned on page 11, lines 13-16.

C: (P18, l1), Authors compare the computing time of their algorithm with the one of Liu et al. (2016) thats uses deep learning. How do both methods compare in terms of performances?

A: Since both models were trained and tested on different datasets, containing data on different geographical regions, the performance of both models cannot be directly compared.

C: For the sake of reproducibility, it would be nice to at least provide in supplementary materials, details about the actual implementation of the methods. For instance, the programming language used, the potential external softwares/packages/libraries used and for which step of the method.

A: We provide relevant details about the actual implementation of the method in supplementary materials of the revised manuscript. The TDA algorithm is implemented in C++ and is compatible with TECA software (Prabhat et al., 2015). The SVM model was imported from Python scikit-learn module. All code is available at `https://github.com/muszyna25/AR-Detection-Method-TDA-ML`. The datasets used in the analysis are available at `http://portal.nersc.gov/project/m1517/cascade/doi/GMD_2018/GMD_2018.html`.

[revised manuscript text omitted]